# An Inexpensive Incubator for Mammalian Cell Culture Capable of Regulating $O_2$, $CO_2$, and Temperature

**Philip Samokhin, Georgina L. Gardner** [ID]**, Chris Moffatt** [ID] **and Jeffrey A. Stuart** *[ID]

Department of Biological Sciences, Brock University, St. Catharines, ON L2S 3A1, Canada;
psamokhin@brocku.ca (P.S.); gg17ww@brocku.ca (G.L.G.); cmoffatt2@brocku.ca (C.M.)
* Correspondence: jstuart@brocku.ca

**Abstract:** Mammalian cell culture is widely used for discovery and development. Recently, increasing attention has been paid to the importance of maintaining physiologically-relevant conditions in cell culture. Although oxygen level is a particularly important consideration, it is rarely regulated by experimentalists. The atmospheric $O_2$ levels commonly used in cell culture are significantly higher than those experienced by most mammalian cells in vivo, leaving cells susceptible to oxidative damage, senescence, transformation, and otherwise aberrant physiology. A barrier to incorporating $O_2$ regulation into most cell culture workflows has been the expense of investing in new equipment, as the vast majority of laboratory $CO_2$ incubators do not regulate $O_2$. Here, we describe an inexpensive (<CAD 1000), portable and user-friendly $O_2/CO_2$ incubator that can establish and maintain physiological $O_2$, $CO_2$, and temperature values within their physiological ranges. We used an Arduino-based approach to add $O_2$ and $CO_2$ control to a temperature-regulating egg incubator. Our incubator was tested against a commercial laboratory $O_2/CO_2$ incubator. Using Presens OxoDish technology, we demonstrate that at a setpoint value of 5% gas-phase incubator $O_2$, media $O_2$ averaged 5.03 (SD = 0.03) with a range of 4.98–5.09%. MCF7, LNCaP and C2C12 cell lines cultured in the incubator displayed normal morphology, proliferation, and viability. Culture for up to one week produced no contamination. Thus, our incubator provides an inexpensive means of maintaining physioxia in routine mammalian cell culture.

**Keywords:** oxygen; physioxia; hypoxia; cell culture; incubator

## 1. Introduction

There is a growing realization amongst biologists of the importance of maintaining physiologically representative conditions during cell culture [1–6]. Recently, this has culminated in the development of physiologic media such as human plasma-like media (HPLM) [1] and Plasmax [4], which are modeled on the human blood plasma metabolome. However, in addition to maintaining nutrients such as glucose and amino acids within a physiological range, the partial pressure of oxygen ($O_2$) is an important consideration for cell culturists [3].

While $O_2$ constitutes ~20% of the atmosphere, its level ranges from 2 to 6% in most mammalian tissues [3]. In the absence of $O_2$ regulation in typical cell culture incubators however, it equilibrates to approximately 18–19%, which is substantially hyperoxic and drives the production of excess reactive oxygen species (ROS) and reactive nitrogen species (RNS) from a variety of cellular sources ([7]; reviewed in [6]). These excess ROS/RNS can induce the pathological oxidation of proteins, lipids and nucleic acids, causing the disruption of a variety of biological processes and signal transduction pathways [8–10].

While consensus is growing around the need to maintain physiological oxygen levels (physioxia) in cell culture, there are barriers to its incorporation into a standard cell culture workflow. The vast majority of existing commercial $CO_2$ incubators currently in use do not regulate $O_2$. An important issue with replacing these incubators is cost; a commercial

$O_2/CO_2$ incubator suitable for mammalian cell culture will cost over CAD 10,000. To lower this barrier, we designed a "homemade" $O_2/CO_2$ incubator for mammalian cell culture that can be assembled from readily available materials for less than CAD 1000. This prototype incubator comprised an Arduino-based system capable of monitoring and regulating $O_2$ and $CO_2$ levels via $N_2$ and $CO_2$ gas injection into an inexpensive, temperature-regulating Styrofoam egg incubator. It is capable of maintaining pre-set $O_2$, $CO_2$ and temperature values within the physiological ranges for most mammalian cells. Although other "lab-assembled" $O_2$ monitoring systems [11], imaging systems [12], and cell culture incubators [13,14] exist, we are unaware of any that simultaneously regulate $O_2$ and $CO_2$ to enable experiments to be conducted at physiological $O_2$ levels. Thus, our incubator represents the least expensive solution to the important issue of maintaining physioxia in cell culture that is currently available.

## 2. Materials and Methods

### 2.1. Materials

The Little Giant Circulated Air Incubator 12300 (Cat.# 12300) was purchased from Miller Manufacturing (Eagan, MN, USA). The Arduino Uno R3 ATMEGA328P EVAL (Cat.# A000066), Gravity: I2C Oxygen Sensor (Cat.# TPX00050), Grove-Button(P) (Cat.# C000142), and Grove-4 pin Male to Grove 4 pin cable (Cat.# C000133), were purchased from Arduino (Arduino.cc). The ExplorIR®-W 60% $CO_2$ Sensor (Cat.# GC-0007) was purchased from $CO_2$ meter.com (2 January 2021; Ormond Beach, FL, USA). The Vipmoon Dupont Jumper Wire was obtained from AJIBI (Calgary, AB, Canada). The Large Solderless Breadboard (Cat.# MB-102), 1N4001 Diode-1 A 50 V (Cat.# 1N4001), TIP120 Darlington Transistor (Cat.# TIP120), 0.25 Watt Resistor (1.5 K $\Omega$), 12 V 1 A Power Supply, 9 V 1000 mA Power Supply (Cat.# SAW12F-090-1000U), Breadboard Friendly DC Barrel Jack (Cat.# PJ-102AH) and Solenoid Valve 12 V 0.25 inch NPT (Cat.# ROB-050) were purchased from BCRobotics (Nanaimo, BC, Canada). The Grove-4-Digit Display (Cat.# RB-See-230), was purchased from RobotShop Inc. (Mirabel, QC, Canada). An Omega HH509R (Cat.# HH509R) from Omega Engineering (St-Eustache, QC, Canada) was used as a secondary thermometer. SDR SensorDish® Reader and OxoDish® were purchased from PreSens Presison Sensing (Regensburg, Germany). Cell lines were purchased from American Type Culture Collection (Manassas, VA, USA). Trypan Blue was obtained from BioShop (Burlington, ON, Canada). Fetal bovine serum (Cat.# F1051), penicillin/streptomycin solution, 0.25% trypsin/EDTA solution were obtained from Sigma-Aldrich (St. Louis, MO, USA). Tissue culture dishes (100 × 20 mm) were obtained from Corning Inc (Corning, NY, USA). Glass Bottom MatTek Dishes (35 mm) were purchased from Mattek Corporation (Ashland, MA, USA). Plasmax media was made in-house using constituents as in (Vande Voorde et al., 2019). $\alpha$-Aminobutyrate (L-2-Aminobutyric acid; Cat.# 438371), L-carnosine (Cat.# 535080) and DL-3-Hydroxybutyric acid sodium salt (Cat.# A 11613-06) were purchased from CEDARLANE (Burlington, ON, Canada). 2-Hydroxybutyrate (2-Hydroxybutyric acid sodium salt; Cat.# S509425) and Ammonium Metavanadate (Cat.# A634095) were purchased from Toronto Research Chemicals Inc. (Toronto, ON, Canada). Unless otherwise stated, all other chemicals, reagents, and solutions (mainly used to make Plasmax) were purchased from Sigma-Aldrich (St. Louis, MO, USA), BioSHOP (Burlington, ON, Canada), or Fisher Scientific (Mississauga, ON, Canada). Pipettes and pipette tips (P1000, P200, P20) were purchased from Thermo Fisher Scientific (Waltham, MA, USA).

### 2.2. Methods

#### 2.2.1. Incubator Design

The 30 cm × 30 cm × 20 cm Little Giant 12300 incubator (Miller Manufacturing, Glencoe, MN, USA) is made of 1 cm thick polystyrene and has temperature regulation capability between ambient room temperature and 40 °C (Supplementary Data: Figure S1A). Two ~10 cm × ~20 cm clear plastic windows in the incubator lid were removed to create space to

house the gas regulation apparatus, which consisted of 3D-printed polyethylene terephthalate glycol (PETG) plastic plates containing holes for sensors and gas tubes (printed using a Prusa i3 MK3S+ 3D printer, Cat.# PRI-MK3S-COM-ORG-PEI, Prusa Research, Prague, Czechia). PETG was obtained from filaments.ca (Cat.# 10-B-4-H-3565, Mississauga, ON, Canada). The Arduino Uno (Uno) microcontroller was used as the main processor for regulating both gases.

The schematics (Supplementary Data: Figure S1B,C) show the pin placements and the circuit layout. For the initial development of the incubator, we used a removable pin breadboard to allow convenient connection and disconnection of components. For the final version (Figure 1), the breadboard was replaced with a Gikfun Solder-able Breadboard Gold Plated Finish Proto Board (Cat.# Bdiergthf-975; DongGuan, GuangDong 523000, China). Components were soldered using 60–40 rosin core solder and a X-Tronic Model #3020-XTS-ST soldering iron (Lincoln, NE, USA). The final version of the case was designed using ®Autodesk Fusion360 software (Mill Valley, CA, USA) and printed using a Prusa MK3S+ 3D printer obtained from Prusa Research a.s. (17000 Prague 7. Czech Republic). The material used was PLActive (SKU: 26-C-3-D-3388), a copper based antimicrobial plastic produced by Copper3D (filaments.ca; Mississauga, ON, Canada).

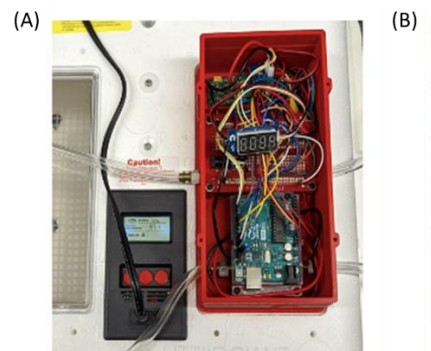
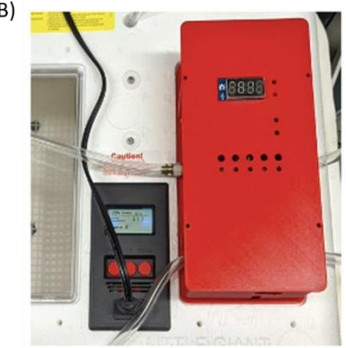
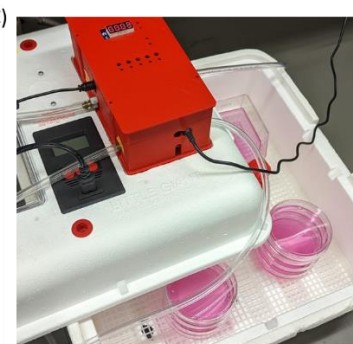

**Figure 1.** $O_2/CO_2$ regulator housing both gas sensors, Arduino controller, solenoid valves, and all associated electronics. (**A**) Showing internal wiring within the 3D-printed box. (**B**) Showing completed and closed unit. (**C**) Showing inside, with stacks of 60 mm tissue culture dishes and T25 tissue culture flask. The incubator can easily hold 25 100 mm tissue culture dishes. More information is available at https://github.com/StuartLab/Incubators (accessed on 15 February 2022).

The Arduino code for running the prototype incubator is presented in Supplementary File S1 and is available, along with details of hardware design including 3-D printing, at https://github.com/StuartLab (accessed on 15 February 2022). Both gas levels are monitored at pre-set intervals and compared to setpoint values. These setpoints can be adjusted in the code as needed. As our goal was to establish physioxia, we used 5% $O_2$, along with the standard 5% $CO_2$. To establish the $O_2$ setpoint below atmospheric, the incubator is flushed with $N_2$. To establish $CO_2$ setpoint above atmospheric, $CO_2$ is brought into the incubator. Gas entry is controlled by dedicated solenoid valves for $N_2$ and $CO_2$. The opening time of the valves is longer when the difference between actual and setpoint values is greater. The LCD display toggles between showing $O_2$ and $CO_2$ values. Pressing the incorporated button pauses gas control for 30 s to allow the incubator lid to be removed without instigating compensatory gas flow.

2.2.2. Validation of Temperature, $O_2$ and $CO_2$ Regulation Performance

The built-in temperature control performance of the incubator was validated with an Omega HH509R handheld digital thermometer (St-Eustache, QC, Canada). Temperature readings were recorded every 20 s for 6 min, with the handheld thermometer placed either immediately beside the prototype incubator's temperature sensor, or at the bottom rack of the incubator. $O_2$ regulation (in the absence of $CO_2$ regulation) at a setpoint of 5% $O_2$ and temperature of 37 °C was measured by taking $O_2$ readings every 2.25 s for 25 min.

$CO_2$ regulation (in the absence of $O_2$ regulation) was measured at a setpoint of 5% $CO_2$ and temperature of 37 °C, with $CO_2$ readings collected every 1 s for 25 min. Simultaneous regulation of 5% $O_2$ and 5% $CO_2$ at a temperature of 37 °C was measured by recording $O_2$ and $CO_2$ readings every 4.25 s for 25 min. All gas measurements were taken following a stabilization period, with readings only taken when the solenoids were closed. To compare the performance of our incubator to that of a commercial-grade incubator, the $O_2$, $CO_2$ and temperature sensors from our incubator were placed inside a Forma Series II Water-Jacketed $O_2/CO_2$ regulating incubator (ThermoFisher, Waltham, MA, USA) set to 5% $O_2$, 5% $CO_2$ and 37 °C. Measurements were recorded every 4.25 s for 25 min following stabilization.

Pericellular $O_2$ fluctuation was measured using a 24-well Presens Oxodish and SDS SensorDish Reader (Regensburg, Germany). Four wells of the Oxodish were loaded with 0.967 mL of medium and placed into our incubator overnight to equilibrate. Media $O_2$ levels were then measured for 3 h.

The rate at which our incubator achieved gas and temperature setpoints of 5% $O_2$, 5% $CO_2$ and 37 °C from room conditions (atmospheric $O_2$ and $CO_2$, 21 °C) was manually measured until setpoint conditions stabilized. Humidity was passively maintained under these conditions using a water reservoir placed in the corner of the incubator.

### 2.2.3. Cell Culture

C2C12, LNCaP and MCF7 cell lines were cultured in Plasmax media supplemented with 2.5% FBS and penicillin (50 I.U./mL)/streptomycin (50 μg/mL). Cells were maintained within a humidified 5% $O_2/CO_2$ atmosphere at 37 °C inside either the prototype incubator, or a standard Forma Series II Water-Jacketed $O_2/CO_2$ regulating incubator (ThermoFisher, Waltham, MA, USA). Prior to placing plates within the prototype incubator, the inside was wiped down with 70% ethanol. Given the size of the incubator, this could be done within the Biological Safety Cabinet. To ensure this measure sufficiently mitigated contamination risks, a contamination test was performed. Three 100 mm culture plates containing 10 mL of Plasmax were placed within both our prototype incubator, and our standard $O_2/CO_2$ incubator, for 7 days. Following this period, plates were imaged using a Zeiss Axio Observer 7 inverted microscope (Oberkochen, Germany) to evaluate whether bacterial or fungal contamination had occurred.

### 2.2.4. Measurement of Pericellular $O_2$ Levels

24-well disposable Presens Oxodish plates and a SDS SensorDish Reader were also used to measure pericellular oxygen levels for cells cultured within the inexpensive "home-made" incubator under 5% $O_2$. The number of cells seeded and the final medium volume in each well were adjusted to mimic the cell density and media column height of the experiments in 10 cm culture plates. Pericellular $O_2$ levels were measured at 10 min intervals for 40 h. Four wells were seeded per cell line, along with four wells containing only media.

### 2.2.5. Cell Viability and Imaging

Cells were seeded on 35 mm Mattek Dishes at densities appropriate for the growth rate of each cell line (75,000–150,000 cells). Three plates were seeded per cell line and incubator type. Media was refreshed after 24 h for all plates. At 48 h, brightfield images of plates were taken using the Zeiss Axio Observer 7 microscope (Oberkochen, Germany). A trypan blue exclusion test of cell viability was then performed. Plates were trypsinized and cells were isolated and resuspended in PBS. Then, 50 uL of cell suspension and 50 uL of 0.4% trypan blue were mixed and allowed to incubate at room temperature for 3 min. Next, 10 uL of the mixture was then added to a hemocytometer and the unstained (viable) and stained (nonviable) cells were counted. Viability percentages and growth rates were determined from these counts.

## 3. Results

The prototype incubator's temperature control performance was validated by comparing temperature measurements made by the incubator's build-in thermal sensor to a secondary handheld thermometer (Omega HH509R). During preliminary testing, we noted that a setpoint temperature of 37.2 °C was required to maintain a temperature of 37 °C as measured immediately beside the incubator's temperature sensor. Thus, with a setpoint of 37.2 °C, the average temperature readings from the built-in and secondary thermometers were 37.05 °C (SD = 0.11) and 37.01 °C (SD = 0.42), respectively. At the same setpoint temperature of 37.2 °C, the average temperature at the bottom rack of the incubator, where cell plates would be located, was 36.85 °C. However, this was in the absence of gas regulation; with periodic puffs of $N_2$ and $CO_2$ at 10 psi, associated with gas regulation, this gradient was not observed (see below). When gases were actively regulated and temperature set to 37.2 °C, the average reading of the incubator thermometer and secondary thermometer was 37.04 °C (SD = 0.11; range 36.8–37.2 °C) and 37.04 °C (SD = 0.32; 36.4–37.5 °C), respectively.

To determine the efficiency of the incubator at regulating $O_2$ and $CO_2$, both independently and concurrently, headspace gas stability was measured. To evaluate $O_2$ regulation, in the absence of $CO_2$, an $O_2$ setpoint of 5% (physioxia) and 37 °C was selected. $O_2$ was measured periodically following setpoint stabilization (Figure 2A), with a mean $O_2$ level of 5.01% (SD = 0.08; range 4.85–5.14) observed over 25 min. Similarly, $CO_2$ regulation, in the absence of $O_2$, was measured at a setpoint of 5% $CO_2$ and 37 °C. The mean $CO_2$ level over 25 min was 5.01% (SD = 0.06; range of 4.85–5.19) (Figure 2B).

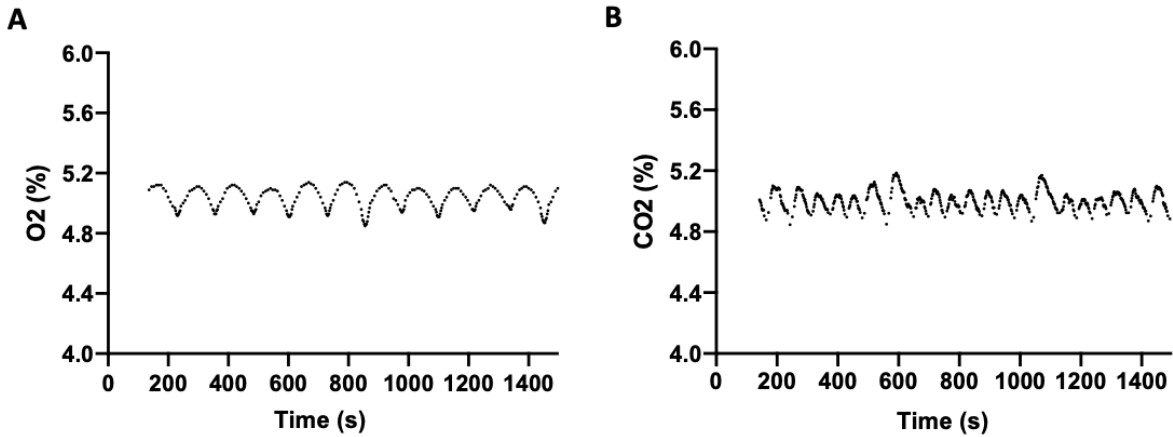

**Figure 2.** $O_2$ and $CO_2$ levels within 'homemade' $O_2/CO_2$ incubator with setpoints of (**A**) 5% $O_2$ or (**B**) 5% $CO_2$ and 37 °C. Only the gas of interest was regulated to 5% in either measurement to assess independent gas regulation. The incubator was allowed to equilibrate prior to measurements.

When all three parameters were maintained simultaneously at 5% $O_2$, 5% $CO_2$, and 37 °C, the amplitude of fluctuation in gas regulation was slightly increased. Following stabilization, the mean $O_2$ level was 4.99% (SD = 0.11) with a range of 4.79–5.16% (Figure 3A), and the mean $CO_2$ level was 4.99% (SD = 0.14) with a range of 4.68–5.26% (Figure 3B). Temperature averaged 37.06 °C (SD = 0.61).

To compare our prototype incubator's performance with a commercial-grade laboratory $O_2/CO_2$ incubator, we collected comparable data from a Forma Series II Water-Jacketed $CO_2/O_2$ set to 5% $O_2$, 5% $CO_2$, and 37 °C. Temperature, $CO_2$ and $O_2$ sensors from the prototype incubator were used, with measurements recorded from sensors placed on both the bottom and top shelf. The average temperature reading was 36.8 °C (SD = 0.30), with no difference observed between sensor position. The average $O_2$ level was 5.01% (SD = 0.02) and $CO_2$ was 5.00% (SD = 0.01). Thus, while our prototype incubator was capable of maintaining gas levels at 5%, it allowed greater fluctuation of gas levels around the setpoint than the commercial-grade Forma incubator.

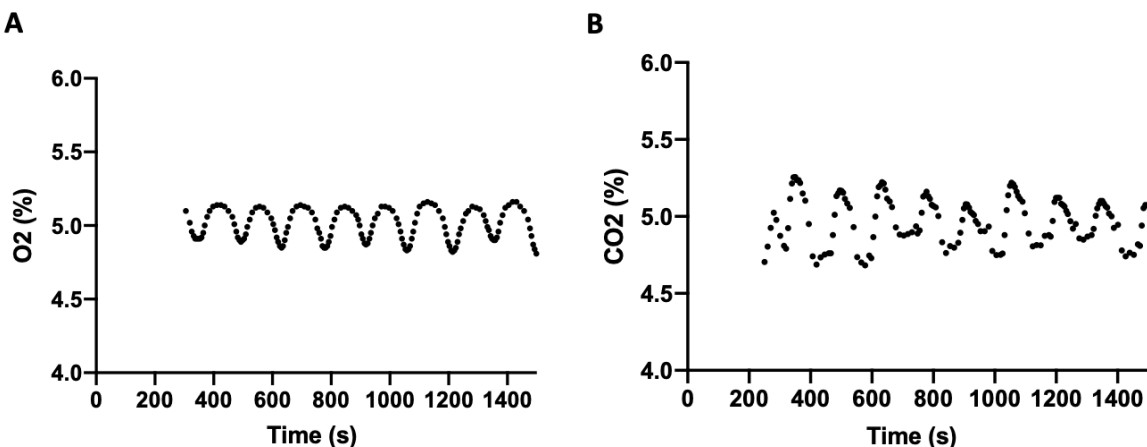

**Figure 3.** (**A**) $O_2$ and (**B**) $CO_2$ levels within 'homemade' $O_2/CO_2$ incubator with setpoints of 5% $O_2$, 5% $CO_2$, and 37 °C. All three variables were regulated simultaneously. The incubator was allowed to equilibrate prior to measurements.

Greater variation in gas levels observed in our incubator may be acceptable for culturing cells. Since $O_2$ equilibration across the water (media) boundary is relatively slow [15], it is unlikely that the vacillation of $O_2$ or $CO_2$ levels in our incubator (compared to the commercial incubator) would be reflected in the cell culture media. We tested this assumption using a PreSens OxoDish® and SDR SensorDish® Reader to monitor medium $O_2$. The OxoDish/SensorDish system uses calibrated, $O_2$-sensitive, fluorescent dots adhered to the tissue culture dish plastic to measure pericellular $O_2$ levels in media immediately adjacent to adherent cells. Following an acclimation period, media $O_2$ levels were measured over 3 h and showed an average of 5.03% (SD = 0.03) with a range of 4.98–5.09% (Figure 4). Notably, this is a substantially narrower range than observed for $O_2$ in the gas phase, and we consider this acceptable variation for cells in culture.

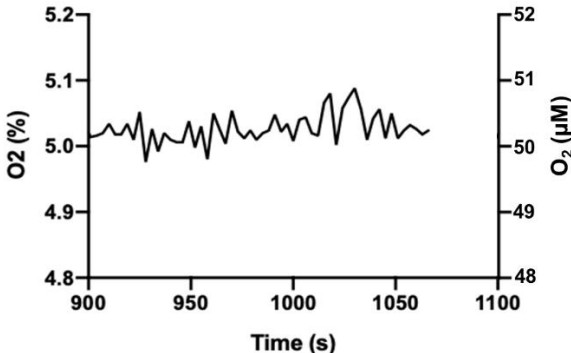

**Figure 4.** Maintenance of $O_2$ levels in media. Data are means of $O_2$ levels measured in four wells of a 24-well Oxodish using a Presens SDR Sensordish. The dish with media was allowed to equilibrate in the incubator overnight and then measurements were collected every 3 min for 3 h.

Since our incubator is smaller than many commercial $O_2/CO_2$ incubators, an additional advantage of its use is the ability of the incubator to reach gas and temperature setpoints relatively rapidly. We demonstrated this by measuring the time taken to reach setpoint conditions (5% $O_2$, 5% $CO_2$) from ambient room conditions (atmospheric $O_2$ and $CO_2$). For both $O_2$ and $CO_2$, setpoints were reached within 5 min. In comparison, ~1.5 h is required to reach similar setpoints in our commercial $O_2/CO_2$ incubator.

As a method of contamination mitigation, the inside of the prototype incubator was wiped down with 70% ethanol prior to use. To determine whether this was a sufficient measure, plates containing Plasmax (2.5% FBS, 1% pen/strep) were left undisturbed in both our incubator and a commercial incubator set to 5% $O_2$, 5% $CO_2$ and 37 °C for 7 days.

Visualization of the plates on day 7 confirmed no contamination occurred in either incubator. Furthermore, plates containing seeded cells cultured for 48 h under the same conditions also saw no signs of contamination, validating our selected disinfecting technique.

To assess the performance of our incubator for cell culture, we next compared cells maintained in our incubator versus the commercial $CO_2/O_2$ incubator over 48 h. MCF7, LNCaP and C2C12 cell lines showed no obvious differences in morphology between either incubator (Figure 5). Similarly, doubling times and cell viability between the incubators were not significantly different. The average doubling times for MCF7, LNCaP and C2C12 cell lines in our incubator were 20.93 h (SD = 1.36), 68.8 h (SD = 0.712) and 16.58 h (SD = 1.04), respectively. In the standard incubator, the doubling times were 9.1 h (SD = 0.211), 68.9 h (SD = 2.65), and 17.0 h (SD = 1.35), respectively. Cell viability was over 90% in all trials, regardless of which incubator was used.

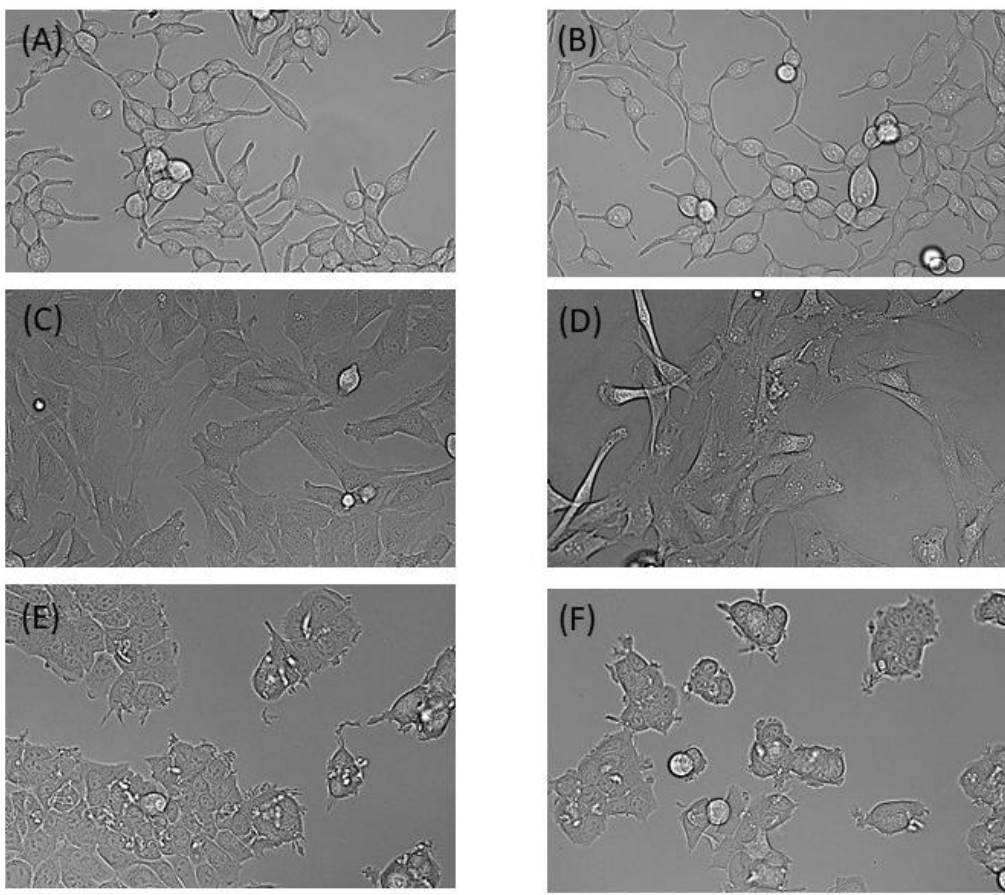

**Figure 5.** LNCaP, C2C12, and MCF7 cell lines cultured for 48 h at 5% $O_2$, 5% $CO_2$ and 37 °C in our incubator (**A,C,E**) versus a commercial laboratory $O_2/CO_2$ incubator (**B,D,F**). (**A,B**) LNCaP cells; (**C,D**) C2C12 cells; (**E,F**) MCF-7 cells. Imaged using a Zeiss Observer 7 inverted microscope (200×).

Pericellular oxygen depletion associated with cellular respiration is a significant consideration when culturing cells in physioxia. Pericellular hypoxia or anoxia is possible depending on the cell density and cellular respiration rates of the cell line [7]. Using the Presens OxoDish/SensorDish system, we show that no pericellular hypoxia occurred in the MCF7, LNCaP and C2C12 cells cultured in our prototype incubator (Figure 6). Average pericellular oxygen for each of these cell lines were 4.55% (SD = 0.156), 4.37% (SD = 0.198), and 4.50% (SD = 0.169), respectively, over 40 h.

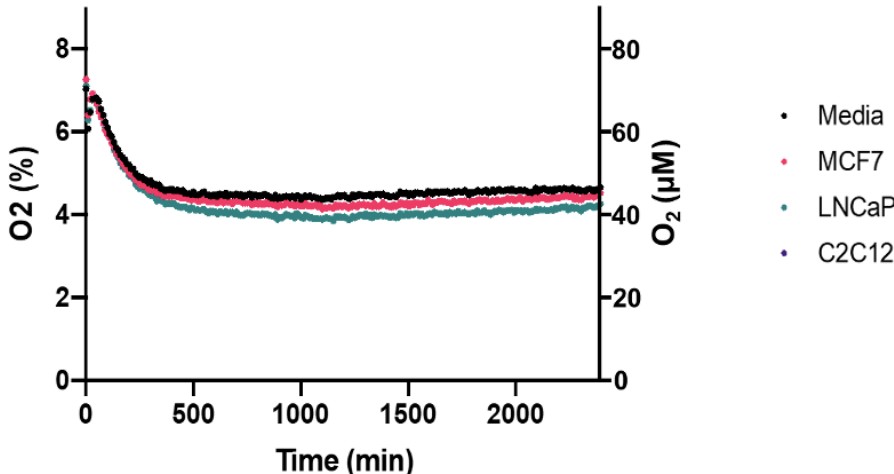

**Figure 6.** Presens Oxodish pericellular oxygen readings for MCF7, LNCaP, C2C12, and control media in 'Homemade' Incubator regulated at 5% $O_2$, 5% $CO_2$ and 37 °C. Average pericellular oxygen observed to stay above 4.3% $O_2$ in all cell lines following gas stabilization.

## 4. Discussion

Our goal was to develop an inexpensive incubation system that cell culturists could use to maintain physiological $O_2$ levels in addition to $CO_2$ and temperature in cell culture. This will provide a means to overcome a current limitation in the field, i.e., the inability to control $O_2$ in mammalian cell culture due to the lack of appropriate equipment. It will facilitate conducting experiments in an environment that more closely resembles physiological conditions.

The system we developed uses a chicken egg incubator with built-in temperature regulation that was fitted with Arduino-controlled $O_2$ and $CO_2$ sensors driving solenoid valves for $CO_2$ and $N_2$ gases. This incubator was effective at maintaining $O_2$ within the physiological range (e.g., 5%) indefinitely, while simultaneously maintaining 5% $CO_2$ and 37 °C. The fluctuation around the setpoint for both $O_2$ and $CO_2$ was greater than we observed in our Forma Series II Water-Jacketed $CO_2/O_2$ incubator. However, the $O_2$ fluctuation was, as expected given the slow equilibration of $O_2$ across the gas-liquid interface [15], significantly dampened in the liquid phase (media). The incubator requires only several minutes to reach setpoints of 5% $O_2$, 5% $CO_2$ from an ambient air starting condition.

At the time of publication, the material cost of our prototype incubator was under CAD 1000. It weighs approximately one kilogram and as such is also highly portable within the lab. Certainly, it is not as portable as the $CO_2$-alone incubators developed by Willbrand et al. [13] or Arumagam et al. [14]. These researchers focused on developing cell incubation systems suitable for fieldwork and/or shipping live cells. Our goal was primarily to provide an inexpensive means to maintain physioxia, and, as such, our incubator is, as currently developed, unsuitable for fieldwork. However, it can be powered by 9 V and 12 V batteries, so if fitted with small $N_2$ and $CO_2$ tanks it could be used for that purpose. Of course, or $O_2/CO_2$ sensing and regulating circuitry could be implemented in a wide range of contexts, including microscope stage gas regulation

Since our incubator is relatively small (30 cm × 30 cm × 20 cm), it also has sufficient portability to facilitate its use in, for example, a biosafety cabinet. This is useful to address another aspect of cell culture under physioxia: the transient exposure to atmosphere during passaging and/or media transfers. We have used the prototype incubator in a 6′ biosafety cabinet as a means to keep our cell cultures in a physiologically appropriate environment during our standard workflow. Since setpoint gas levels are reached within minutes of opening and closing the incubator lid to access the cultures, we are able to avoid temperature and gas shocks to the cells.

## 5. Conclusions

The incubator we describe here will provide an affordable solution to the critical problem of maintaining physioxia in cell culture. These are key considerations for any cell culturist wishing to maintain a more physiological environment in vitro.

**Supplementary Materials:** The following supporting information can be downloaded at: https://www.mdpi.com/article/10.3390/oxygen2010003/s1.

**Author Contributions:** Conceptualization, J.A.S.; software, P.S.; formal analysis, P.S., G.L.G. and C.M.; investigation, P.S., G.L.G. and C.M.; resources, J.A.S.; writing—additional draft preparation, P.S. and G.L.G.; writing—review and editing G.L.G. and J.A.S.; supervision, J.A.S.; project administration, J.A.S.; funding acquisition, J.A.S. All authors have read and agreed to the published version of the manuscript.

**Funding:** This work was funded the Natural Sciences and Engineering Research Council (NSERC) of Canada. Discovery Grant RGPIN–2020-05274.

**Institutional Review Board Statement:** This study did not require Research Ethics Board approval.

**Informed Consent Statement:** Not applicable.

**Data Availability Statement:** Data can be requested from jstuart@brocku.ca.

**Conflicts of Interest:** The authors declare no conflict of interest.

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
