# Peer review of "An Inexpensive Incubator for Mammalian Cell Culture Capable of Regulating O2, CO2, and Temperature"

_oxygen, doi:10.3390/oxygen2010003_

Round 1

Reviewer 1 Report

The manuscript which title is “An inexpensive incubator for mammalian cell culture capable 2 of regulating O2, CO2, and temperature” is interesting. The incubator could provide inexpensive mechanism in the study of physioxia condition in the cell culture. However, the cellular damage could not find in the figure 5. The authors should be provided the reason in the results. And the authors provide the results of 5% O2, 5% CO2, 5%O2+CO2 in the results of cell culture. More the authors should provide large field of microscope and relative survival rate in the results.    

Author Response

Reviewer #1

The manuscript which title is “An inexpensive incubator for mammalian cell culture capable 2 of regulating O2, CO2, and temperature” is interesting. The incubator could provide inexpensive mechanism in the study of physioxia condition in the cell culture. However, the cellular damage could not find in the figure 5. The authors should be provided the reason in the results. And the authors provide the results of 5% O2, 5% CO2, 5%O2+CO2 in the results of cell culture. More the authors should provide large field of microscope and relative survival rate in the results.   

Figure 5 is not showing cellular damage. Rather, it is showing the absence of obvious differences in cell morphology when cells are grown in the prototype incubator versus the commercial laboratory incubator. There is no evidence of cell damage or death in our prototype incubator. The population doubling time data corroborate this finding – the cells grow the same in either incubator.

Reviewer 2 Report

Dear Authors, I have been doing cell culture since more than 35 years. I am a biologist by training. I understand certain aspects of your introductory statements, but I have enormous difficulties in following what this "egg incubator" looks like and why it could work for a standard laboratory used to handle petridishes, T-flasks, Rollerbottles etc. A simple drawing and/or photo as Fig. 1. could help. The second issue is the volume of cell culture to be done. The culture is static, ie. the oxygen transfer to the cells is occuring by diffusion. Experimental cell culture needs to do in series to explore variations of the culture media and/or other conditions. How many different culture vessels can be entered into your egg incubator? Can they by stacked on top of each other? 

I understand that you reduce the oxygen level in the incubator by adding nitrogen into the flow of gas. your culture data are shown only for a rather short period of time. I.e. not more than several days. You have to provide data on several subcultivations, expansions of cells etc. possibly at least over 30-50 population doublings. 

In other words, while I trust your expertise in the coding and setting up of the equipment, I do not see (yet) how this system can be made available to a larger cell culture audience. 

I therefor reject this paper for now, but encourage you to make it more explanatory and readable to a wider audience. 

Using a term "Arduino-based" without any explanation what this stands for does not help to encourage further reading. the Materials and Method section is full of non-related / or irrelevant information for a cell culture audience. This can be shortened and the emphasis should be provided towards the general applicability of the system. 

It should be noted also that this system does not apply for suspension cultures, which have become very popular in research. So it is only applicable for adherent cultures, probably mostly primary animal/human derived cells, cell strains and immortalized cells (all adherent).

Author Response

Reviewer #2

Dear Authors, I have been doing cell culture since more than 35 years. I am a biologist by training. I understand certain aspects of your introductory statements, but I have enormous difficulties in following what this "egg incubator" looks like and why it could work for a standard laboratory used to handle petridishes, T-flasks, Rollerbottles etc. A simple drawing and/or photo as Fig. 1. could help. The second issue is the volume of cell culture to be done. The culture is static, ie. the oxygen transfer to the cells is occuring by diffusion. Experimental cell culture needs to do in series to explore variations of the culture media and/or other conditions. How many different culture vessels can be entered into your egg incubator? Can they by stacked on top of each other?

We have provided a photo (Figure 1C) of the interior of the incubator, showing several tissue culture dishes and flasks inside. This prototype incubator is intended to approximate the conditions inside a typical laboratory CO2 incubator, though it is smaller than many commercially available versions. The plates can be stacked as high as the 18cm internal height permits. As now is mentioned in the figure 1 legend, at least 25 100mm TC plates can be comfortably fit (this is 5 stacks of 5. More could fit, but it would be awkward to move them).

I understand that you reduce the oxygen level in the incubator by adding nitrogen into the flow of gas. your culture data are shown only for a rather short period of time. I.e. not more than several days. You have to provide data on several subcultivations, expansions of cells etc. possibly at least over 30-50 population doublings.

Commercial CO2 incubators with O2 regulation capability also use N2 to expel excess atmospheric O2. Our approach is the same as this, just must less expensive. 30-50 populations would require about one year to complete. We believe any information gained from this would not be worth the time or effort. There is no reason to believe that there would be an effect of time.

In other words, while I trust your expertise in the coding and setting up of the equipment, I do not see (yet) how this system can be made available to a larger cell culture audience.

We have placed detailed instructions on how to make the incubator on our laboratory GitHub site. We have other cell biology-related projects available on this site and they are being used actively. Our mitochondrial network analysis tool (MiNA), for example, is being used by hundreds if not thousands of mitochondrial biologists worldwide. Users are downloading the code from our GitHub. We hope the same will be true for our incubator. However, we would also be willing to help assemble them for people, at least initially, at cost. Our goal is to provide tools, primarily for academic research labs, to do cell culture in physioxia and we think this is attainable.

Using a term "Arduino-based" without any explanation what this stands for does not help to encourage further reading. the Materials and Method section is full of non-related / or irrelevant information for a cell culture audience. This can be shortened and the emphasis should be provided towards the general applicability of the system.

Arduino is basically Lego electronics – cross compatible components that can be used to build just about anything. Many DIY scientists use Arduino components in their labs to construct specific tools. If a reader is put off by the term ‘Arduin-based’, they are probably unlikely to be interested in making their own incubator using our instructions in any case. But, those individuals could contact us and we would help them.

It should be noted also that this system does not apply for suspension cultures, which have become very popular in research. So it is only applicable for adherent cultures, probably mostly primary animal/human derived cells, cell strains and immortalized cells (all adherent).

This system DOES apply for suspension cultures. The incubator is too small, however, to accommodate most roller-culture systems. But, in my experience there are many thousands of academic labs using cell culture in dishes that could make use of our incubator.

Reviewer 3 Report

I appreciate the idea to search the solutions in order to allow the groups that are not "rich" to do the research. However, this approach can be applied only for the research which is out of the regulatory field i.e. only for the basic research and not for development of cell products for clinical studies and cell production production for clinical studies themselves (certifications needed).

I have one comment in order to improve the presentation of data: In adition to % it would be useful to give the dissolved O2 concentrations in µM (Figures 2,3,4 and 5). First, it is more "natural" since it is about dissolved O2, and, second, it would allow to directly compare the results withother studies.

Author Response

Reviewer #3

I appreciate the idea to search the solutions in order to allow the groups that are not "rich" to do the research. However, this approach can be applied only for the research which is out of the regulatory field i.e. only for the basic research and not for development of cell products for clinical studies and cell production for clinical studies themselves (certifications needed).

We agree that our incubator is unlikely to be of interest to a commercial pharmaceutical company. It is a tool that is likely to be valued by researchers at universities who are trying to do much with limited funding.

Round 2

Reviewer 2 Report

the authors essentially disregarded this reviewers comment... they are in a world which seems outside of standard cell culture approaches. I understand the argument of low budget, but that does not mean that they have to present the data in a poorly understandable form and refuse to even extent their work for a few more weeks, to show reproducibility and longer term performance. 

Unless the authors are ready to engage in a more serious effort to make this manuscript more accessible to a wider audience, I continue to reject it. 

Suspension culture is not "roller bottle" culture. The authors seem not to be aware that an entire industry is based on suspension cultures, from the milliliter scale to thousands of litres , provided by stirring or shaking. 

This manuscript is a resubmission of an earlier submission. The following is a list of the peer review reports and author responses from that submission.

Round 1

Reviewer 1 Report

The authors report an inexpensive ‘homemade’ incubator for mammalian cell culture capable of regulating O2, CO2, and temperature. This seems to be an interesting bespoke development and has the potential to be replicated by others for rapid bench-top effortless culturing. The reported incubator is rather very small, portable and weighing <1 kg and has been integrated with sensors and electromechanical valves that are controlled with low-cost Arduino microcontroller. Maintaining physiologically relevant stable environmental conditions all through tissue/cell cultures becomes more and more important every day in life science R&D especially in personalised medicine, disease modelling and drug screening. Therefore, we need active research to establish reliable and smart environmental control and monitoring systems and all works in that direction are highly appreciated. The manuscript is well drafted in terms of appropriate English usages and understandability.  

However, overall, this is a weak manuscript. Specifically, at many places the manuscript reads like an undergraduate lab report. It lacks sufficient rigor and the structuring and formatting required for a scientific paper but rather, contains unnecessary electronics detailing, scattered uncompiled plots, figures with dissimilar time scaling and limited scientific soundness and justifications. The main drawback of the manuscript is that it lacks the description of novelty. Authors must justify the merits of their system against similar systems with appropriate citations. In addition, a manuscript about an alternative incubator for cost-effective cell cultures cannot be accepted without appropriate validation experiments with cells. The manuscript must be drafted in a scientific manner with concise and relevant experimental details validating the system purpose, advantages, limitation and future prospective. In the current form, for e.g. a cardiac cell culturist cannot conclude whether the system can be replicated for cardiac cultures or a neuronal culturist for neuronal cultures.  More specific comments are listed below

In the abstract:  ’other changes’ – what does it mean?

“Here we describe an inexpensive incubator developed from a bird egg incubator capable of regulating temperature by incorporating an Arduino-based system that adds regulation of O2 and CO2” - Confusing sentence

this was not reflected in the cell culture media” – Have you validated this using different types of mediums?

Even though the title contains the terms ‘O2’, ‘CO2’, ‘home-made’ and ‘temperature’, the abstract doesn’t address all of them. One is forced to read the whole manuscript to get an overview of the control features; for. e.g. on how the temperature and humidity are regulated etc. The abstract is poorly drafted and doesn’t summarise the key aspects and does not justify the novelty of the system in contrast to similar ones. 

The introduction also requires substantial redrafting to cover the key advantages, the scientific need, and critical technological comparisons with appropriate citations. Readers must get an overview from the abstract and introduction on what this paper is about, how useful it can be in someone’s research and what exact environmental parameters are regulated (& how and with what accuracy etc.). The only newness stated is the reduction in cost. Reduction in cost alone doesn’t fulfil the necessary requirements of scientific novelty and significance for a journal article. Authors must consider the concepts like innovate research Vs incremental research and carefully handle and draft the contents. In addition, the cost related figures must be included also in the abstract and/or introduction together with other scientific and technological advantages to draw the readers’ interest.

Consider removing the descriptions of the pin connections (e.g. lines 109 to 113,  115 to 120, 121 to 124 etc. ) with appropriate wiring diagrams. Remember this is a scientific article and a not a lab report. You could include design diagrams, PCB diagrams etc. in the supplementary section.

Due to overall power demands’- What does this mean? I cannot understand why the authors include lengthy detailing of 9-12V/power requirements of some parts/components. Replicating engineers will definitely check the data sheets of all components. So, inclusion of individual power requirements of a couple of parts that particularly do not add anything pivotal to the main story can be avoided.

How did you calibrate your sensors and what are their accuracies? How do you control the humidity and evaporation in long-term cultures? Why didn’t you include humidity sensors? Are your sensors suitable for a highly humid incubator environment?

Why do you see a regular periodicity in the signal in Fig 2 A and B?

Instead of several scattered plots, combine plots (e.g. Fig 2 A and B as a single plot with left and right axes).

The measurement times in various plots are notably different. Why is it so?

The gas solubility of the medium is temperature dependent. Have you thought about it while doing measurements for. e.g. in Fig.4.?

Though the incubator volume is ~ 18 k cm3, it still takes ~ 1 hour for O2 and CO2 stabilization which is somewhat comparable to the stabilization times of a conventional cell culture incubator! Why is it so? Can you reduce the stabilization time by some gas control tweaks or tricks?

It is weird to see different time scales in Fig 5. Do combine the 3 plots into a single plot. It looks like the temperature hasn’t stabilized in Fig5 A? Please re-scale the y axis between ~20  -40.

How do you sterilize the incubator? Will the sensor or electronics addons create sterilization challenges?

It is a cell culture incubator. But no cell culture experiments together with data on cell viability, cell proliferation etc are included. That is a must!

Some important citations are missing. Include citations to similar bespoke incubators and comparable systems and show the comparable merits.

You may consider the below points while drafting novelty considerations in the next version.

Demonstrate the suitability of your system not only for normoxia but also for highly regulated hypoxia experimentations. Good quality hypoxia systems  e.g. for cardiac hypoxia studies may attract a more wider readership

Consider measuring pH of the medium in real time for a couple of days (with or without cells) to unambiguously prove that the system regulates gas levels in a more biological perspective. A stable pH is a reliable outcome of a stable gas control (and temperature stability) which will be much more directly convincing to the biologists about the system stability and suitability.

Starting from the abstract, clearly list other novelties such as: Portability, small-form factor, low weight, bench-top, easy to assemble, requires very low gas volumes only, potentially low stabilization time, integration of additional sensors, possibility of smart control, may be also smart phone compatibility via Arduino channels etc.

Moreover, before editing your manuscript ‘read read read read’ some similar articles carefully. Finally, sorry to say but I will be recommending to reject the manuscript this time, but good luck with the next submission with the aforementioned changes.

Reviewer 2 Report

Manuscript: oxygen-1404660

"An inexpensive ‘homemade’ incubator for mammalian cell culture capable of regulating O2, CO2, and temperature" by Samokhin et al.

The authors described a low-cost incubator for regulating temperature, O2, and CO2, developed from a bird egg incubator by incorporating an Arduino-based system. The system might be able to regulate the O2 concentration of mammalian cell cultures within the physiological range.

Point of critique:

The authors state that their system can be an important tool in the context of mammalian cell cultivation. While they demonstrate the basic functionality of their system, they do not show any results concerning the short- and long-term cultivation of the corresponding mammalian cells. At least initial pilot experiments demonstrating the applicability of the adjustable incubator in the context of cell cultures would be desirable and would add authenticity to the article.

Without appropriate experiments, the authors should rephrase their statements regarding actual applicability in cell culture experiments.

Taken together, the manuscript might be acceptable for publication after a minor revision.